# Minimally Invasive Distal Pancreatectomy Techniques: A Contemporary Analysis Exploring Trends, Similarities, and Differences to Open Surgery

**DOI:** 10.3390/cancers14225625

**Published:** 2022-11-16

**Authors:** Fernanda Romero-Hernandez, Sarah Mohamedaly, Phoebe Miller, Natalie Rodriguez, Lucia Calthorpe, Patricia C. Conroy, Amir Ashraf Ganjouei, Kenzo Hirose, Ajay V. Maker, Eric Nakakura, Carlos Corvera, Kimberly S. Kirkwood, Adnan Alseidi, Mohamed A. Adam

**Affiliations:** 1Department of Surgery, University of California, San Francisco, CA 94143, USA; 2Department of Surgery, Division of Surgical Oncology, University of California, San Francisco, CA 94143, USA

**Keywords:** distal pancreatectomy, minimally invasive surgery, hand-assisted laparoscopic, total laparoscopy

## Abstract

**Simple Summary:**

While the adoption of minimally invasive pancreatectomy had lagged, it has become more mainstream in recent years. Hand-assisted laparoscopic technique, an adjunct to laparoscopic surgery may offer the benefits of a total laparoscopic approach while mitigating the technical challenges associated with it. Previous studies investigating the different approaches in distal pancreatectomy have predominately focused on comparing outcomes between total laparoscopic (LDP) vs. open distal pancreatectomy (ODP); however, limited research has focused on outcomes associated with the hand-assisted distal pancreatectomy (HALDP) approach. This study demonstrates that compared to ODP, LDP was associated with improved surgical site infection rates. There was no difference in surgical site infection rates between ODP and HALDP. LDP was associated with longer operative times (+10 min only). Surgeon comfort and experience should decide the operative approach, but it is important to discuss the differences between these approaches with patients.

**Abstract:**

Limited contemporary data has compared similarities and differences between total laparoscopic (LDP), hand-assisted (HALDP), and open distal pancreatectomy (ODP). This study aimed to examine similarities and differences in outcomes between these three approaches in a contemporary cohort. Methods: Patients undergoing elective LDP, HALDP, and ODP in the NSQIP dataset (2014–2019) were included. Descriptive statistics and multivariate regression analyses were employed to compare postoperative outcomes. Results: Among 5636 patients, 33.9% underwent LDP, 13.1% HALDP, and 52.9% ODP. Compared with the LDP approach, surgical site infections were more frequent in HALDP and ODP approaches (1.2% vs. 2.6% vs. 2.8%, respectively, *p* < 0.01). After adjustment, the LDP approach was associated with a significantly lower likelihood of surgical site infection (OR 0.25, *p* = 0.03) when compared to ODP. There was no difference in the likelihood of surgical site infection when HALDP was compared to ODP (OR 0.59, *p* = 0.40). Unadjusted operative times were similar between approaches (LDP = 192 min, HALDP = 193 min, ODP = 191 min, *p* = 0.59). After adjustment, the LDP approach had a longer operative time (+10.3 min, *p* = 0.04) compared to ODP. There was no difference in the adjusted operative time between HALDP and ODP approaches (+5.4 min, *p* = 0.80). Conclusions: Compared to ODP, LDP was associated with improved surgical site infection rates and slightly longer operative times. There was no difference in surgical site infection rates between ODP and HALDP. Surgeon comfort and experience should decide the operative approach, but it is important to discuss the differences between these approaches with patients.

## 1. Introduction

Minimally invasive surgery (MIS) techniques have been widely adopted across multiple surgical fields and are now considered the standard of care for many procedures, such as laparoscopic cholecystectomy and appendectomy [1,2]. However, the adoption of MIS for pancreatectomy has lagged. One explanation could be the technical difficulty of approaching the pancreas laparoscopically, due to its retroperitoneal location alongside major vascular structures [3]. However, recent studies have shown favorable results regarding the short-term benefits and safety of laparoscopic distal pancreatectomy, including lower surgical site infection rates, reduced blood loss, and shorter length of hospital stay, while maintaining comparable efficacy to the open approach (ODP) [3,4,5].

The hand-assisted laparoscopic technique, an adjunct to laparoscopic surgery, may offer the benefits of a total laparoscopic approach while mitigating the technical challenges associated with it. Hand-assisted surgery is performed through the placement of a hand-port to allow the insertion of the surgeon’s hand into the peritoneal cavity [6]. Hand-assisted techniques allow for improved tactile sensation and distance perception and increase the ease with which surgical specimens are removed from the abdominal cavity [7,8,9]. Furthermore, it may allow faster response in the event of unexpected bleeding and facilitate suture ligation, which could explain the reduction in intraoperative blood loss and operative times reported in some studies [8,9,10]. However, large population-based studies comparing minimally invasive approaches in the setting of colectomy have demonstrated an increased risk of surgical site infection with hand-assisted colectomy, challenging the premise that hand-assisted surgery carries the same risks as a total laparoscopic approach [10,11,12,13].

Previous studies investigating the different surgical techniques in distal pancreatectomy have predominately focused on comparing outcomes between total laparoscopic (LDP) vs. open distal pancreatectomy (ODP); however, limited research has focused on outcomes associated with the hand-assisted distal pancreatectomy (HALDP) approach [13,14], especially in the modern era where minimally invasive distal pancreatectomy techniques have become more accepted, beyond the early learning curve. Given the lack of comparisons between these approaches and prior evidence suggesting minimal differences in operative times [10] and increased overall risk of surgical site infection in the setting of colectomy [10,12,13], the objective of this study was to examine the surgical site infection rates and operative times across LDP, HALDP, and ODP for distal pancreatectomy utilizing a large national database.

## 2. Materials and Methods

### 2.1. Data Source and Study Cohort

The American College of Surgeons (ACS) National Surgical Quality Improvement Program (NSQIP) database was queried to identify all patients who underwent elective minimally invasive (HALDP and LDP) and open distal pancreatectomy (ODP) (CPT codes 48140 and 48146) from 2014 to 2019 [15]. Patients younger than 18 years old and who had preoperative jaundice or biliary stenting, ascites, preoperative infection (superficial/deep surgical site infection, abscess, pneumonia, or urinary tract infection), were ventilator-dependent, on hemodialysis, or who underwent vascular resection were excluded. To ensure that the outcomes of interest were only driven by the distal pancreatectomy, patients who underwent other major concurrent or additional surgical procedures during the same general anesthetic event as the distal pancreatectomy were excluded from the analysis. Variables extracted from the database included: patient age, sex, race, ethnicity, diagnosis, pathological stage (T, N, and M stages), neoadjuvant therapy, baseline comorbidities, preoperative albumin levels, operative time, gland texture, postoperative complications, length of stay, need for reoperation, 30-day readmission, and 30-day mortality. Due to the de-identified nature of the dataset, this study was granted exemption status by our Institutional Review Board.

### 2.2. Primary and Secondary Endpoints

The primary endpoints were the occurrence of surgical site infection and operative time. Secondary endpoints included composite overall complications, composite significant complications, clinically relevant (B/C) postoperative pancreatic fistula formation, postoperative transfusion, discharge to facility, and length of hospital stay. A composite of overall complications was created and included infectious (deep wound infection and dehiscence, abscess, urinary tract infection, post-operative sepsis, and/or post-operative septic shock), pulmonary (pneumonia, reintubation and/or ventilator wean failure), cardiovascular (deep vein thrombosis, pulmonary embolism, stroke, cardiac arrest, and/or myocardial infarction) and renal complications (acute renal failure), post-operative blood transfusion, reoperation, 30-day readmission, and 30-day mortality. Urinary tract infection and acute renal failure were excluded from the overall complications composite to create the significant complications composite variable.

### 2.3. Statistical Analysis

Descriptive statistics for patients’ demographic, preoperative, and intraoperative characteristics and postoperative outcomes were compared between patients undergoing LDP, HALDP, and ODP surgery. For continuous variables, the nonparametric Kruskal–Wallis test was used and for categorical variables, the χ^2^ test was employed.

The association between each covariate (patient age, sex, race, ethnicity, neoadjuvant therapy, baseline comorbidities (diabetes, smoking, dyspnea, functional status, COPD, ascites, CHF, hypertension requiring medication, bleeding disorder), body mass index, ASA class, preoperative albumin levels, unintentional preoperative weight loss, wound classification, pathological diagnosis and stage (T and N stages), and gland texture) and the outcomes of interest was first tested using univariate analysis. Then, variables having a significant association (*p*-value < 0.2) with the primary/secondary outcomes of the univariate analyses were entered into a multivariate model. Each final model included covariates with a *p*-value less than 0.05 using a stepwise backward elimination method.

Linear regression models were used to determine the adjusted differences in operative time and length of hospital stay between surgical approaches with adjustment for patient age, race, diabetes mellitus, hypertension, smoking, congestive heart failure, unintentional preoperative weight loss, ASA class, body mass index, neoadjuvant treatments, wound classification, pathological diagnosis, and soft gland texture. Results are presented in the form of adjusted odd ratios (aOR) and mean ratios, including 95% confidence intervals (CI) with a *p*-value < 0.05 for significance. Statistical analyses were performed using SAS version 9.4 (SAS Institute Inc., Cary, NC, USA).

## 3. Results

A total of 5636 patients who underwent elective distal pancreatectomies between 2014 and 2019 were identified. Of these, 33.9% (n = 1913) underwent LDP, 13.2% (n = 741) HALDP, and 52.9% (n = 2982) ODP. Baseline demographics across the three approaches are shown in Table 1. Patient age was similar for LDP, HALDP and ODP approaches, and most patients were females (*p* = 0.002) (Table 1). Use of the different approaches remained the same between 2014 to 2019: LDP (from 32.9% to 33.2%); HALDP (13.8% to 11.7%); ODP (53.4% to 55.1%); *p* = 0.249 (Figure 1).

Patients who underwent ODP were more likely to be diabetic (26.2%, *p* < 0.001), have a higher ASA class (66.5%, *p* < 0.001), have more preoperative weight loss (7.0%, *p* < 0.001), history of bleeding disorder (3.8%, *p* = 0.017) and higher clinical stage (Table 1). There were no differences in the other comorbidities such as smoking status, preoperative steroid use, COPD, and malnourishment between the different approaches. Patients who underwent LDP and HALDP more often had pancreatic cysts compared to patients who underwent ODP (LDP 29.6% vs. HALDP 27.5% vs. ODP 22.5%, *p* < 0.01). Adenocarcinoma was more common in patients who underwent ODP (LDP 18.5% vs. HALDP 26.6% vs. ODP 40.2%, *p* < 0.01).

### 3.1. Surgical Site Infection

In unadjusted analysis, compared with the LDP approach, surgical site infection rates were more frequent in the HALDP and ODP approaches (1.2% vs. 2.6% vs. 2.8%, respectively, *p* = 0.01) (Table 2). After adjustment for confounders, the LDP approach was associated with a significantly lower likelihood of surgical site infection (aOR 0.25, 95% CI 0.07–0.84, *p* = 0.03) when compared to ODP. There was no difference in the likelihood of surgical site infection when HALDP was compared to the ODP approach (aOR 0.59, 95% CI 0.17–0.03, *p* = 0.40) (Table 3).

### 3.2. Operative Times

In unadjusted analysis, operative times were similar between patients who underwent LDP, HALDP, and ODP approaches (LDP 192 min vs. HALDP 193 min vs. ODP 191 min, *p* = 0.59). However, after adjustment for confounders, the LDP approach showed slightly longer operative times than ODP (+10 min; 95% CI 0.53–19.99, *p* = 0.04). There was no difference in the adjusted operative time when HALDP was compared to ODP (+5.4 min; 95% CI −7.88–18.70, *p* = 0.80).

### 3.3. Secondary Endpoints

Overall and significant complication rates differed between approaches (overall complications LDP 24.3% vs. HALDP 23.9% vs. ODP 31.6%, *p* < 0.01; significant complications LDP 17.6% vs. HALDP 17.7% vs. ODP 25.3%, *p* < 0.01) (Table 2). After adjustment for confounders, LDP and HALDP approaches were associated with decreased odds of overall complications when compared to ODP LDP aOR 0.66, 95% CI 0.50–0.87, *p* < 0.01; HALDP aOR 0.56, 95% CI 0.38–0.84, *p* < 0.01). Similarly, both LDP and HALDP approaches were associated with decreased odds of significant complications compared to ODP (LDP OR 0.63, CI 0.46–0.85, *p* < 0.01; HALDP OR 0.65, 95% CI 0.42–0.99, *p* < 0.05, respectively) (Table 3). There was no difference in the odds of developing clinically-relevant (B/C) postoperative pancreatic fistula after LDP or HALDP compared to ODP (LDP aOR 1.28, 95% CI 0.93–1.78, *p* = 0.13; HALDP aOR 0.71, 95% CI 0.42–1.22, *p* = 0.22). LDP and HALDP approaches were also associated with a decreased likelihood of blood transfusion compared to ODP (LDP aOR 0.40, 95% CI 0.22–0.74, *p* < 0.01; HALDP aOR 0.40, 95% CI 0.17–0.98, *p* < 0.05) (Table 3).

Length of hospital stay was shorter in the LDP approach when compared to HALDP and ODP (LDP 4 days vs. HALDP 5 days vs. OPD 6 days; *p* < 0.01) (Table 2). After adjustment for confounders, both LDP (−0.26, 95% CI −0.31–−0.22, *p* < 0.01) and HALDP (−0.26, 95% CI −0.32–−0.20, *p* < 0.01) were associated with a significantly shorter length of hospital stay when compared to ODP. There was no difference in readmission rates between approaches (LDP 14.1% vs. HALDP 15.7% vs. ODP 15%, *p* = 0.55).

## 4. Discussion

This large study explores similarities and differences between surgical approaches for distal pancreatectomy in the modern era. We found that the total laparoscopic approach was associated with the lowest risk of surgical site infection compared with either the hand-assisted or open approaches. After adjustment, the association between the total laparoscopic approach and the lower odds of surgical site infection remained statistically significant. No difference in the odds of surgical site infection between the hand-assisted and open distal pancreatectomy approaches was found. Adjusted operative times were longer in the total laparoscopic approach by ten minutes. Both total laparoscopic and hand-assisted approaches were associated with lower odds of overall and significant complications, and shorter lengths of hospital stay. Despite these similarities and differences, the use of any of the three approaches remained the same over the past six years.

Previous studies have been inconsistent regarding surgical site infection rates between minimally invasive and open approaches in distal pancreatectomy. A meta-analysis from 2011 by Venkat et al. that included 733 laparoscopic and 1041 open distal pancreatectomies, demonstrated a lower risk of surgical site infection in the laparoscopic group (OR 0.45) [5]. In contrast, a recent randomized control trial that compared 51 laparoscopic vs. 57 open distal pancreatectomies found no differences in surgical site infection rates between approaches [16]. However, neither of these studies considered or included patients who underwent hand-assisted distal pancreatectomy. For example, Venkat et al.’s meta-analysis study defined the laparoscopic group as patients who underwent distal pancreatectomy with and without hand assistance, making no further distinction between approaches in their analysis. Moreover, the randomized control trial by de Rooiji et al. only included laparoscopic and robotic approaches [16]. Our study found a significant association between surgical approach and surgical site infection. In particular, the total laparoscopic approach had a lower risk of surgical site infection, but the hand-assisted approach had a similar risk to open distal pancreatectomy. Therefore, it is important to make the distinction between the total laparoscopic and the hand-assisted approaches when it comes to surgical site infection. Another difference in our study is that we were able to use a contemporary cohort and adjust for important confounders that could have potentially influenced the development of surgical site infection, such as wound classification and soft pancreatic gland texture [17].

The underlying mechanism of the association between distal pancreatectomy approaches and superficial surgical site infection may be related to the increased length of the incision required in the hand-assisted technique [11]. The hand-assisted approach requires about a 6–8 cm incision in the upper middle line to place the hand-port [14,16], which is typically longer than the incision needed for specimen extraction in the total laparoscopic approach. The continuous pressure exerted between the hand port and the abdominal wall, and the frequent reintroductions of the surgeon’s hand could increase tissue trauma and ischemia, making the surgical site more prone to infections [12,17,18,19,20,21].

Surgical site infection is associated with significant morbidity and cost, especially after pancreatectomy [19,22,23]. Surgical site infection also contributes to an increased burden on the healthcare system; on average, surgical site infection increases hospital costs by an average of $11,462 after major hepatopancreatobiliary surgeries [21]. Although anecdotally considered innocuous compared to other potentially devastating complications after pancreatectomy, such as grade C postoperative pancreatic fistula, surgical site infection is clinically-relevant and should be mitigated to improve patient outcomes and minimize burdens on the healthcare system [17,21].

Our study found that the total laparoscopic approach was associated with increased operative times. While the difference between the total laparoscopic and open groups is statistically significant, an argument can be made that the 10 min operative time difference may not be clinically relevant. Similar to our results, the LEOPARD trial demonstrated longer operative times in patients who underwent total laparoscopic distal pancreatectomy when compared to the open (217 vs. 179 min) [16]. In another study, Gamboa et al. reported their experiences on the short-term outcomes of 433 patients who underwent minimally invasive and open distal pancreatectomy at two medical centers [14]. However, the majority (89%) underwent pure laparoscopic distal pancreatectomy and only a small number of patients (11%) underwent hand-assisted distal pancreatectomy. Their unadjusted analysis found shorter operative times in the hand-assisted group than in the total laparoscopic group (−24 min). While we found similar results, the study by Gamboa et al. did not adjust for confounders that may have impacted surgical time such as comorbidities, pathological diagnosis, and vascular resections. Our results add the advantage of demonstrating the findings pertaining to the hand-assisted approach with accounting for confounders.

Both univariate and multivariable analyses showed lower overall complications and significant complications in the total laparoscopic and hand-assisted approaches when compared to open, which are consistent with previous studies [5,14]. However, in a retrospective analysis that included 1667 patients and compared patient outcomes by surgical approach in mucinous cystic neoplasms, no differences were found in complication rates [13]. Specifically, this study that compared 46 patients who underwent hand-assisted vs. 76 patients who underwent total laparoscopic or robotic approaches found no differences in postoperative and major complications. The authors also compared the hand-assisted approach vs. 153 patients who underwent open distal pancreatectomy and did not find statistical differences in postoperative and major complications. However, unlike the present study, their analysis was limited both by a small sample and the absence of adjustment for confounders.

Our study is limited by its retrospective nature. NSQIP does not include data on why a certain surgical approach was chosen. NSQIP only includes postoperative outcomes within 30 days; there is no data available on long-term outcomes including the development of an incisional hernia. NSQIP does not include data on surgeon or hospital volume nor type of facility (i.e., academic vs. community), which may be associated with different surgical approaches. Additionally, CPT codes used in NSQIP do not specify who underwent spleen-preserving distal pancreatectomy and who did not. However, the use of NSQIP data allows for the analysis of a large sample size which includes diverse institutions and surgeons, therefore reducing overall bias. Additionally, our study provides a unique perspective by analyzing surgical outcomes by the type of minimally invasive approach. Previous studies have typically grouped the two approaches under one “minimally invasive” group. By separating the hand-assisted and total laparoscopic approaches, we were able to provide more granular results on the outcomes of interest.

Our findings have important implications for clinical and future research. Our results suggest that while the hand-assisted approach shares similarities with the total laparoscopic approach, there are also important differences, such as the rates of surgical site infection. HALDP could be used as an alternative option to convert from the total laparoscopic approach in cases where dissection is difficult, unclear anatomy or the risk of hemorrhage is high. It may also be used in cases of malignancy, where HALDP facilitates palpation of tumor and staging and can facilitate dissection for voluminous lesions [24]. Future research should consider separately analyzing patients who underwent hand-assisted distal pancreatectomy from those who had total laparoscopic distal pancreatectomy as they may confer different outcomes. While surgeon comfort and experience should decide the operative approach, there are cases that may be more suitable for the hand-assisted approach such as those with expected technical difficulty or dissection and thus cannot be completed laparoscopically. The hand-assisted approach is still associated with better outcomes compared to the open approach.

## 5. Conclusions

Compared to open distal pancreatectomy, total laparoscopic distal pancreatectomy is associated with improved rates of surgical site infection. However, there was no difference in surgical site infection rates between open and hand-assisted distal pancreatectomy. Total laparoscopic distal pancreatectomy is associated with a slightly longer operative time (+10 min only). Surgeon comfort and experience should decide the operative approach, but it is important to discuss the differences between these approaches with patients. Further studies are needed to evaluate the direct benefits of hand-assist, particularly in the setting of conversion from laparoscopic as well as a postoperative incisional hernia.

## Figures and Tables

**Figure 1 cancers-14-05625-f001:**
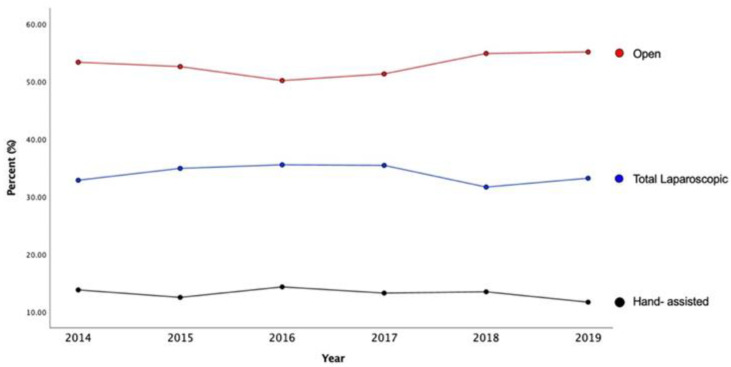
Time Trends of All Patients Undergoing Distal Pancreatectomy by Surgical Approach.

**Table 1 cancers-14-05625-t001:** Demographic and Preoperative Clinical Characteristics of Patients Undergoing Distal Pancreatectomy in NSQIP (2014–2019).

	Open (n = 2982)	Hand-assisted (n = 741)	Total Laparoscopic (n = 1913)	*p* Value
**Age** (years)				0.374
<65	1520 (51.0 %)	397 (53.6%)	1021 (53.4%)	
65–80	1302 (43.7%)	312 (42.1%)	794 (41.5%)	
>80	160 (5.4%)	32 (4.3%)	98 (5.1%)	
**Female sex**	1675 (56.2%)	413 (55.7%)	1166 (61.0%)	0.002
**Race**				<0.001
White	2262 (75.9%)	578 (78.0%)	1326 (69.3%)	
Black	272 (9.1%)	69 (9.3%)	158 (8.3%)	
Asian	91 (3.1%)	22 (3.0%)	113 (5.9%)	
Other	14 (0.5%)	3 (0.4%)	13 (0.7%)	
**Hispanic**	130 (4.4%)	41 (5.5%)	95 (5.0%)	0.317
**BMI** (kg/m^2^, Median, IQR)	27.63 (24.17–31.78)	28.58 (24.79–33.3)	28.21 (24.27–32.6)	<0.001
**ASA Class**				<0.001
1	29 (1.0%)	10 (1.4%)	32 (1.7%)	
2	829 (27.8%)	234 (31.6%)	662 (34.6%)	
3	1984 (66.5%)	466 (62.9%)	1135 (59.3%)	
4	136 (4.6%)	30 (4.1%)	82 (4.3%)	
**Diabetes Mellitus**				<0.001
No	2201 (73.8%)	566 (76.4%)	1515 (79.2%)	
Non–Insulin–Dependent	337 (11.3%)	63 (8.5%)	123 (6.4%)	
Insulin–Dependent	444 (14.9%)	112 (15.1%)	275 (14.4%)	
**Smoking**	482 (16.2%)	103 (13.9%)	266 (13.9%)	0.061
**Dyspnea**	167 (5.6%)	41 (5.5%)	118 (6.2%)	0.674
**COPD**	121 (4.1%)	28 (3.8%)	91 (4.8%)	0.391
**CHF**	15 (0.5%)	7 (0.9%)	2 (0.1%)	0.008
**Hypertension requiring medications**	1484 (49.8%)	400 (54.0%)	926 (48.4%)	0.036
**Steroids Use**	115 (3.9%)	38 (5.1%)	74 (3.9%)	0.263
**Weight loss (** **≥10%)**	210 (7.0%)	38 (5.1%)	57 (3.0%)	<0.001
**Bleeding Disorder**	113 (3.8%)	21 (2.8%)	45 (2.4%)	0.017
**Preoperative Albumin** (g/dL, Median, IQR)	4.1 (3.9–4.4)	4.1 (3.8–4.4)	4.2 (3.9–4.4)	0.003
**Pathological** **Diagnosis**				<0.001
Adenocarcinoma	1198 (40.2%)	197 (26.6%)	354 (18.5%)	
Neuroendocrine Tumor	516 (17.3%)	204 (27.5%)	567 (29.6%)	
Pancreatitis	211 (7.1%)	25 (3.4%)	63 (3.3%)	
Pancreatic Cyst	670 (22.5%)	224 (30.2%)	703 (36.6%)	
**Neoadjuvant Chemotherapy**	407 (13.7%)	29 (3.9%)	60 (3.1%)	<0.001
**T Stage**				<0.001
Not Applicable	1121 (37.6%)	322 (43.5%)	967 (50.6%)	
T1	373 (12.5%)	124 (16.7%)	302 (15.8%)	
T2	550 (18.4%)	129 (17.4%)	331 (17.3%)	
T3	813 (27.3%)	148 (20.0%)	275 (14.4%)	
T4	36 (1.2%)	1 (0.1%)	5 (0.3%)	
**N Stage**				<0.001
Not Applicable	993 (33.3%)	272 (36.7%)	603 (31.5%)	
N0	693 (23.2%)	110 (14.8%)	220 (11.5%)	
N1	1121 (37.6%)	323 (43.6%)	970 (50.7%)	
**Wound Classification**				<0.001
Clean	433 (14.5%)	88 (11.9%)	362 (18.9%)	
Clean–Contaminated	2346 (78.7%)	631 (85.2%)	1488 (77.8%)	
Contaminated	187 (6.3%)	21 (2.8%)	59 (3.1%)	
Dirty/Infected	16 (0.5%)	1 (0.1%)	4 (0.2%)	
**Soft Pancreas**	859 (28.8%)	147 (19.8%)	443 (23.2%)	<0.001

Abbreviations: NSQIP: American College of Surgeons National Surgical Quality Improvement; IQR: Interquartile Range; BMI: Body Mass Index; COPD: Chronic Obstructive Pulmonary Disease; CHF: Congestive Heart Failure. Kruskal–Wallis test was used for continuous variables and χ^2^ test for categorical variables.

**Table 2 cancers-14-05625-t002:** Distribution of Outcomes in Patients Undergoing Distal Pancreatectomy by Surgical Approach.

	Open (n = 2982)	Hand-Assisted (n = 741)	Total Laparoscopic (n = 1913)	*p* Value
**Overall Complications ***	943 (31.6%)	177 (23.9%)	465 (24.3%)	**<0.001**
**Significant** **Complications ^+^**	753 (25.3%)	131 (17.7%)	337 (17.6%)	**<0.001**
**CR–POPF**	372 (12.5%)	119 (16.1%)	290 (15.2%)	**0.003**
**Blood Transfusion**	275 (9.2%)	19 (2.6%)	53 (2.8%)	**<0.001**
**Discharge to Facility**	138 (4.6%)	36 (4.9%)	53 (2.3%)	**0.002**
**Surgical Site Infection**	82 (2.8%)	19 (2.6%)	23 (1.2%)	**0.001**
**Operative Time** (min, Median, IQR)	191 (139–255)	193 (150–246)	192 (149–244)	0.585
**LOS** (days, Median, IQR)	6 (5–7)	5 (4–6)	4 (4–6)	**<0.001**

Abbreviations: IQR: Interquartile Range; Min: Minutes; CR-POPF: Clinically Relevant (Grade B/C) Postoperative Pancreatic Fistula; LOS: Length of Stay. * Overall complications composite included infectious (deep wound infection and dehiscence, abscess, urinary tract infection, post-operative sepsis, post-operative septic shock), pulmonary (pneumonia, reintubation, and ventilator wean failure), cardiovascular (deep vein thrombosis, pulmonary embolism, stroke, cardiac arrest, myocardial infarction) and renal complications (acute kidney failure), post-operative blood transfusion, reoperation, 30-day readmission, and 30-day mortality. ^+^ Significant complications composite included infectious (deep wound infection and dehiscence, abscess, post-operative sepsis, post-operative septic shock), pulmonary (pneumonia, reintubation, and ventilator wean failure), cardiovascular (deep vein thrombosis, pulmonary embolism, stroke, cardiac arrest, myocardial infarction), post-operative blood transfusion, reoperation, 30-day readmission, and 30-day mortality. Kruskal–Wallis test was used for continuous variables and χ^2^ test for categorical variables.

**Table 3 cancers-14-05625-t003:** Multivariable Logistic Regression with backward elimination of Patients Undergoing Hand-Assisted or Total Laparoscopic Compared to Open Distal Pancreatectomy.

	aOR	95% CI	*p*-Value
**Surgical Site Infection**			
Open	Reference		
Hand-assisted	0.59	0.17–2.03	0.403
Total Laparoscopic	0.25	0.07–0.84	**0.025**
**Overall Complications**			
Open	Reference		
Hand-assisted	0.56	0.38–0.84	**0.005**
Total Laparoscopic	0.66	0.50–0.87	**0.003**
**Significant Complications**			
Open	Reference		
Hand-assisted	0.65	0.42–0.99	**0.046**
Total Laparoscopic	0.63	0.46–0.85	**0.003**
**CR-POPF**			
Open	Reference		
Hand-assisted	0.71	0.42–1.22	0.215
Total Laparoscopic	1.28	0.93–1.78	0.133
**Transfusion**			
Open	Reference		
Hand-assisted	0.40	0.17–0.98	**0.045**
Total Laparoscopic	0.40	0.22–0.74	**0.003**
**Discharge to Facility**			
Open	Reference		
Hand-assisted	0.80	0.36–1.78	0.579
Total Laparoscopic	0.44	0.23–0.84	**0.013**

CR-POPF: Clinically Relevant (Grade B/C) Postoperative Pancreatic Fistula, aOR: Adjusted Odds Ratio, CI: Confidence Interval. Adjusting for age, sex, race, diabetes mellitus, hypertension, smoking, congestive heart failure, weight loss, wound classification, pathological diagnosis, body mass index, ASA class, and soft gland texture.

## Data Availability

The data that support these findings are housed with the American College of Surgeons. Data are available in a de-identified fashion to participants of the NSQIP Program.

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
