# Peer review of "Minimally Invasive Distal Pancreatectomy Techniques: A Contemporary Analysis Exploring Trends, Similarities, and Differences to Open Surgery"

_cancers, 2022, doi:10.3390/cancers14225625_

Round 1

Reviewer 1 Report

Thank you for the opportunity to review this interesting paper by Romero-Hernandez et al. Limited evidence is available on the use and outcomes of hand-assisted LDP and therefore this paper is important additionally the authors were able to include a large amount of patients. When the groups were compared for baseline characteristics, clear differences between the open, HA and laparoscopic DP group were seen. Showing that there is clear treatment allocation bias. The authors corrected for these differences in additional analyses.

Overall, the manuscript is well-written and interesting to read, I have some suggestions:

Overall suggestions:

-       Consider altering the used abbreviations. TL and HL are not widely used maybe alter to LDP and HALDP?

Methods:

-       Were other types of adjusted analyses considered, for example propensity score or other types of matching (definitely not the holy grail but provides some advantages in some cases)?

-       As mentioned, HA techniques can have an advantage over open, for example in the learning curve. Is there any information available on the type of centers that perform these procedures? Regarding volume / academic? Experience of surgeons?

Discussion:

-       I would suggest to elaborate a little bit more on the advantages of HALDP in the discussion. Because when HAL is compared to LDP the latter has clearly more advantages compared to ODP. In which occasions would you suggest to choose a HAL approach over LDP or ODP? There are probably no studies to reference to regarding this but some of the author’s opinions regarding this are also interesting.

Reviewer 2 Report

The objective of this study was to examine the surgical site infection rates and operative times across TL (totally laparoscopy), HAL (HALS technique) and ODP (open approach) for distal pancreatectomy utilizing a large national database. The results showed a reduction of SSI and hospital stay in laparoscopic option with a quite longer operative time after adjustment for cofounders. HALS did not show differences respect to open approach.

There are several bias in this national database, in particular it not includes data on why a certain surgical approach was chosen and selection biases are inevitably present. In particular, the laparoscopic approach may be related to easier cases whereas open one to difficult ones (this is also showed in the higher rate of neoadjuvant therapy and T3 stage in open approach).

Moreover, the analysis does not include robotic approach that is gaining nowadays increasing popularity and widespread.

Reject

Reviewer 3 Report

I want to thank the authors for the opportunity to review this manuscript.

This manuscript mentioned about the minimally invasive surgery for pancreatic diseases, and was well written for contents and methodology that used big data.  Recently, total laparoscopic approach that not hand-assisted was common procedure and, I felt there were little novelty.  As you mentioned in the manuscript, the backgrounds such as choice of the surgical approach, lymph nodes dissection were ambiguous, and it were weak points of this manuscript. 

Round 2

Reviewer 3 Report

The authors responded to the concerns raised by the reviewers.